# A Case Study of Polyether Ether Ketone (I): Investigating the Thermal and Fire Behavior of a High-Performance Material

**DOI:** 10.3390/polym12081789

**Published:** 2020-08-10

**Authors:** Aditya Ramgobin, Gaëlle Fontaine, Serge Bourbigot

**Affiliations:** CNRS, INRAE, Centrale Lille, UMR 8207—UMET—Unité Matériaux et Transformations, Univ. Lille, F-59000 Lille, France; aditya.ramgobin@gmail.com (A.R.); gaelle.fontaine@univ-lille.fr (G.F.)

**Keywords:** high-performance polymer, polyether ether ketone (PEEK), reaction to fire, thermal stability, decomposition mechanism

## Abstract

The thermal and fire behaviors of a high-performance polymeric material—polyether ether ketone (PEEK) was investigated. The TG plots of PEEK under different oxygen concentrations revealed that the initial step of thermal decomposition does not greatly depend on the oxygen level. However, oxygen concentration plays a major role in the subsequent decomposition steps. In order to understand the thermal decomposition mechanism of PEEK several methods were employed, i.e., pyrolysis–gas chromatography–mass spectrometry (Py–GC–MS), thermogravimetric analysis (TGA) coupled with a Fourier-transform infrared spectrometer (FTIR). It was observed that the initial decomposition step of the material may lead to the release of noncombustible gases and the formation of a highly crosslinked graphite-like carbonaceous structure. Moreover, during the mass loss cone calorimetry test, PEEK has shown excellent charring and fire resistance when it is subjected to an incident heat flux of 50 kW/m². Based on the fire behavior and the identification of pyrolysis gases evolved during the decomposition of PEEK, the enhanced fire resistance of PEEK was assigned to the dilution of the flammable decomposition gases as well as the formation of a protective graphite-like charred structure during its decomposition. Moreover, at 60 kW/m², ignition occurred more quickly. This is because a higher rate of release of decomposition products is achieved at such a heat flux, causing a higher concentration of combustibles, thus an earlier ignition. However, the peak of heat release rate of the material did not exceed 125 kW/m².

## 1. Introduction

For more than half a century, plastics have been increasingly used as alternatives to traditional materials such as wood or metals in various applications. They versatility and ease with which they can be processed make them valuable assets for their use in everyday life.

Despite possessing outstanding properties and numerous applications, most of the polymeric materials (plastics) have one undeniable downside: they are highly flammable. In the event of a fire, most plastics, in their virgin form, degrade and release fuel and toxic gases. This can cause severe material damage as well as human casualties and deaths. In Europe, 4000 fire-related deaths are reported every year. Ninety percent of fires in the European Union happen in buildings and because of the use of an increasing amount of flammable materials. Recent statistics have shown that it takes only three minutes for a fire to involve an entire room [1]. Therefore, most plastics need to be flame retarded in order to be used in most environments.

Advances in polymeric materials have allowed the development of high-performance plastics that resist high temperatures and release only little fuel and are therefore virtually non-flammable. Some of these materials are used in industries whereby extreme conditions are met. Such industries include the aeronautics industry, microelectronics or fireproof overalls. Examples of such materials are polyimide (Kapton^®^), polyether ether ketone (PEEK) or aromatic polyamides such as Kevlar^®^. These materials all exhibit extreme fire resistance and thermal stability [2].

As a result, high performance polymeric organic materials are being increasingly used in a wide array of applications varying from the automobile industry, the construction industry as well as aerospace. These applications require that the materials in question display a high resistance to extreme conditions such as high temperature or fire.

However, in a fire scenario with extreme temperatures, such materials can still decompose, releasing combustibles that can fuel a fire. This decomposition may be due to an incident heat flux or the combination of heat and oxygen. Indeed, there are reports suggesting that the presence of oxygen can play a role in speeding up the decomposition of a material, increasing the rate of released compounds available to eventually feed the fire. This role has long been assumed to be catalytic [3] whereby the degradation temperature of a material is lowered and the rate of degradation is increased. However, other reports have shown that oxygen may, on the contrary, delay the intermediate stage of the thermal decomposition [4,5]. Furthermore, in a fire scenario, the availability of oxygen to the burning materials may vary greatly. In a well-ventilated system, the oxygen concentration may be considered constant, however, in the closed room on fire, the level of oxygen decreases drastically.

While the presence of oxygen shows the tendency of decreasing the decomposition temperature of most polymeric materials, some seem to be unaffected to it. Indeed, some high performance materials such as polyether ether ketone (PEEK) exhibit a thermal decomposition onset at the same temperature be it in air or under inert atmosphere [2]. This makes PEEK a highly interesting material in the sense that the catalytic effect of oxygen is not noticeable when observing solely the decomposition temperatures. It is therefore of interest to attempt at understanding the underlying properties that bring about such a resistance to the oxidation under high temperatures to PEEK.

Therefore, in this study, the influence of oxygen on the thermal decomposition and fire behavior of a high-performance polymeric material (polyether ether ketone, PEEK) was investigated.

One of the most widely method adopted to investigate the thermal stability of the materials was thermogravimetric analysis (TGA) in different atmospheres (atmospheres containing different oxygen levels). This provided insight to the effect that the nature of the surrounding atmosphere has on the modes of decomposition of the materials. Moreover, attempts to elucidate the thermal decomposition pathway of the polymer when it is subjected to an elevated temperature were made. To achieve this, the gases evolved during the thermal decomposition of PEEK were identified via two different methods: pyrolysis–gas chromatography–mass spectrometry (Py–GC–MS) and TGA coupled with a Fourier-transform infrared spectrometer (TGA–FTIR). However, before going into the elaboration of a thermal decomposition pathway, the fire behavior of this material was also studied. Finally, attempts at elucidating the thermal decomposition mechanism were made. This furthered the comprehension of the fire behavior and thermal stability of this material.

## 2. Materials and Methods

### 2.1. Materials

Polyether ether ketone (PEEK) was kindly supplied by Quadrant Engineering Polymers and Plastics (Dagneux, France).

### 2.2. Thermogravimetric Analysis

Thermogravimetric analyses (TGA) were conducted on a Netzsch Libra instrument. Samples of 9–10 mg were placed in open alumina pans and heated under different percentages of oxygen (0%, 2%, 4%, 8%, 12% and 20%) at a heating rate of 10 °C/min up to 900 °C. Samples were ground in liquid nitrogen and placed into open alumina pans.

TGA–FTIR measurements were carried out to study the evolved pyrolysis gases. This was conducted using a TA instruments (TGA Discovery) coupled with a Nicolet iS-10 infrared spectrometer. Samples (~10 mg) were heated in a 250 μL alumina crucible from 50 °C to 800 °C with a heating rate of 20 °C/min under nitrogen atmosphere. A balance purge flow of 15 mL/min and a sample purge flow of 50 mL/min was maintained. A transfer line with an inner diameter of 1 mm was used to connect TGA to the infrared cell. The temperature of the transfer line and the gas cell was kept at 225 °C. Prior to this, samples were kept for 2 h under nitrogen stream. IR spectra were collected in 400–4000-cm^−1^ spectral range (resolution: 4, laser frequency: 15,798.7 cm^−1^).

### 2.3. Simultaneous Thermal Analysis (STA)

Simultaneous thermal analyses were performed on a Netzsch Jupiter instrument. A blank was performed with an empty crucible. The temperature program was as follows: cool down to −30 °C; ramp-up of temperature to 900 °C at a heating rate of 10 °C/min. Heat flow curves and TGA curves were obtained simultaneously.

### 2.4. Py–GC–MS

Samples (approximately 200 µg) were analyzed by pyrolysis GC–MS (Shimadzu, GC–MS-QP2010 SE). GC preparation was carried out with a fused silica capillary column (SLB 5 ms) of 30-m length and 0.25-mm thickness. Analyses were carried out both in direct pyrolysis mode and desorption method. The temperature selection for desorption is based on the TGA program used for the PEEK samples, i.e., with a heating rate of 20 °C/min. The initial column temperature was held at 35 °C for a period that corresponds to desorption time followed by a temperature ramp at 10 °C/min to a final temperature of 300 °C and isotherm for 5 min. For direct pyrolysis, the furnace is set for the degradation temperature (700 °C which corresponds to completion of second major degradation step) and sample is pyrolyzed for 0.5 min. Column oven temperature is programmed in the following way. The initial column temperature was held at 35 °C for 1 min. followed by a temperature ramp at 10 °C min^−1^ to a final temperature of 300 °C and isotherm for 20 min. Helium was used as a carrier gas at pressure of 94.1 kPa with a split ratio of 100. The transfer line was maintained at 280 °C. The MS was operated under electron ionization (EI) mode. An online computer using GC–MS real time analysis and PY-2020i software-controlled GC–MS system. The eluted components were identified by library search and only significant peaks observed in the total ion chromatograms were studied and compared to a mass spectral database (GC–MS postrun analysis and NIST).

This allowed for the separation of the decomposition products that are evolved when PEEK is heated at 20 K/min. The decomposition gases analyzed using Py–GC–MS were coherent with those identified with TGA–FTIR. However, this did not provide a real time analysis of the evolved gases. Indeed, using this technique, all the evolved gases during thermal decomposition are kept at 35 °C the beginning of the chromatography column. It is only after the heating process that the separation of the gases is initiated. The advantage of this method is that it provides the possibility of identifying individual mass spectra per decomposition product.

In order to have a real time analysis of the evolved gases during thermal decomposition, EGA–MS was performed. The sample was left for 1 min @ 50 °C, then heated up at a heating rate of 20 °C/min up to a temperature of 750 °C. The GC column was kept at 300 °C for 42 min. This allowed the real time analysis of the evolved gases during thermal decomposition as they travel straight to the MS analyzer as soon as they are evolved. This allowed for the plotting of the intensity profiles of specific *m/z* values, which correspond to decomposition products that were identified by TGA–FTIR and Py–GC–MS. It also provides insight regarding different steps of the thermal decomposition of PEEK under pyrolytic conditions.

### 2.5. Fire Testing

In order to evaluate the burning behavior of the flame retarded materials, the mass loss cone calorimeter (MLC) was used.

The MLC is a bench-scale reaction to fire test which provides a combustion scenario that is typical of developing or developed fires. The measurements were carried out on a fire testing technology (FTT) mass loss calorimeter device (ISO 13927 [6], ASTM E906 [7]). The schematic representation of the device is shown in Figure 1. The procedure involved exposing plates (100 × 100 × 3.7 mm^3^) in horizontal orientation with heating elements. Samples were wrapped in aluminum foil leaving the upper surface exposed to the heater (external heat flux of 35 kW/m^2^ corresponding to a mild fire scenario or 50 kW/m² and 60 kW/m² corresponding to a well-developed fire scenario) and placed on a ceramic backing board at 35 mm from the cone base.

The MLC measured the temperature of the evolved gases using a thermopile located at the top of the chimney. The calibration of the heat release rate (HRR) was performed with methane. A methane flow of 0 to 6.7 mL/min was burnt above the sample holder to obtain a calibration curve of the heat release as a function of methane flow [8].

The values measured by MLC were the heat release rate (HRR), the peak of heat release rate (pH RR), the total heat release (THR), the time to ignition (TTI) and the mass loss of the sample during combustion (ML). All measurements were performed at least thrice. The presented curves are the worst case of repeatable results. The acceptable error of measurement was estimated at 10% for all values.

To identify the gases released during the burning of the polymeric materials tested, the MLC was coupled with a Fourier-transform infrared using a heated transfer line (to avoid condensation of combustion products).

The gas-picking pistol and transfer line were provided by M&C Tech Group. FTIR, Antaris^TM^ Industrial Gas System, was provided by Thermo Fisher. The transfer line between MLC and FTIR was 2 m long and was heated up to 180 °C. Before analyzing the gases by FTIR, soot particles were filtered by two filters (2 and 0.1 μm) made of glass fibers and ceramic, respectively. The FTIR gas cell was set to 185 °C and 652 Torr. The optical pathway was 2 m long and the chamber of the spectrometer was filled with dry air. FTIR spectra obtained using MLC–FTIR were treated using OMNIC software (laser frequency: 15,798.0 cm^−1^, resolution: 0.500). To quantify gases, spectra need to be recorded at different concentrations for targeted gases and a quantification method needs to be created using TQ Analyst. In creating the method, representative regions in the FTIR spectra of the selected gas needed to be chosen and interactions with other gases needed to be taken into account. Using MLC–FTIR, the following gases could be quantified: water, carbon monoxide, carbon dioxide, nitrogen dioxide and hydrogen cyanide. Quantification was reproducible within 10%.

## 3. Results and Discussion

### 3.1. Thermal Stability

TGA is an efficient method to study the thermal decomposition temperatures of a material using milligram scale samples. By applying a thermal stress on a material in a controlled atmosphere, the decomposition temperature of the sample, for the thermal stress applied, can be deduced by analyzing the temperature at which the onset of mass loss is observed. Under pyrolytic conditions, temperature is the only factor that affects the decomposition of the material.

As mentioned previously, in a fire scenario, the bulk of a burning material undergoes pyrolysis in absence of oxygen. However, while many reports suggest that in a fire scenario, a burning polymeric material is only subjected to pyrolytic thermal stress, the presence of even a small amount of oxygen may influence its decomposition behavior [9] and, as such, on its burning behavior. Therefore, the impact of low oxygen concentrations (0%–12% oxygen) on the decomposition PEEK was studied. Furthermore, even though the oxygen concentration is often considered as limited in the fire scenario in a closed room, this is not the case in a well ventilated one, where oxygen can be brought by convection. Additionally, when a material is far from a flame, it can still be subjected to an incident heat flux. In this case, the material is subjected to thermo-oxidative degradation. It is therefore equally important to evaluate the thermal stability of the material in air.

Therefore, the thermal stability of PEEK was studied under different atmospheres to intricately understand its thermal decomposition behavior under different types of fire scenarios.

The TG plots and their DTG (derivative of the thermogravimetric curve) of PEEK when heated at 10 °C/min under different atmospheres are shown in Figure 2. The data are summarized on Table 1.

Under nitrogen, the temperature at the onset of PEEK thermal decomposition (T_95%_) was evaluated at 574 °C and the maximum mass loss rate (11.0 wt%/min) occurs at 592 °C. A nonzero DTG value is also observed at temperatures ranging from 615 °C to 800 °C corresponding to a mass loss ranging from around 60 wt% at 615 °C to approximately 50 wt% at 800 °C. At higher temperatures, there is little mass loss recorded. Indeed, the final residual mass of PEEK at 900 °C under nitrogen is 48%. This suggests that the product formed during the pyrolysis of PEEK is quite stable under pyrolytic conditions and at this temperature. This also suggests that the pyrolysis of PEEK is at least a two-step process, one occurring at the onset of the decomposition and another that overlaps with the first degradation process. Furthermore, it was reported that the PEEK is a highly charring polymer. This suggests that during the first step of the decomposition, the reaction occurring leads to a thermodynamically stable char structure [10,11]. Following this char-formation step, a second decomposition step is observed. This step corresponds to the slow degradation of the char at high temperatures. It explains the low, albeit nonzero, values on the DTG curve above 650 °C.

With increasing oxygen concentrations, the TG plots show that there is little change in terms of the temperature at the onset of the decomposition. This is evidenced by the temperature at which the sample records 5% mass loss, which ranges from 565 °C (at 4% O_2_) to 575 °C (at 8% O_2_). This suggests that the inertness or oxidative nature of the atmosphere in which PEEK is heated has little effect on the temperature at which it starts to decompose. This is even more deducible by looking at the temperature at which the peak mass loss rate is recorded (589–593 °C). It is therefore likely that this first step of thermal decomposition corresponds to the formation of a charred structure

Furthermore, the TG plots also show that the oxygen concentration has a significant effect on the second part of the decomposition occurring at temperatures above 650 °C. Indeed, after the first decomposition step, whereby the MLR_MAX_ is recorded, there is a second slower decomposition step, which can also be seen on the DTG curves. These DTG curves evidence that for the second stage of the decomposition, a higher mass loss rate is associated with a higher oxygen concentration. From this, it can be deduced that that this decomposition stage corresponds to the oxidation of the charred structure formed in the first step. This would explain the oxygen concentration dependence of the mass loss rate where oxygen is a limiting factor in the thermo-oxidative decomposition of the char formed in the first step.

Moreover, it is interesting to note that at 900 °C, only the TG plots of PEEK under nitrogen, and 2% oxygen show some residual mass. This means that the charred structure formed during the initial thermal decomposition stage is completely oxidized at 900 °C even at oxygen concentrations as low as 4% while in inert atmospheres, the residual structure corresponds to almost half of the initial mass of the material.

This means that in certain fire scenarios, especially those where the material is subjected to pyrolytic stress, flaming PEEK can form a char which would act as a thermal and physical barrier. This char can limit the access of heat to the bulk of the material as well as prevent further decomposition products from leaving the bulk to reach the flame to feed the fire.

While the key temperatures relating to the thermal stability and decomposition behavior of PEEK under inert atmosphere were extracted using TGA, enthalpic data concerning the thermal behavior of PEEK under nitrogen were evaluated using STA (Figure 3).

From the DSC curve (Figure 3), a small exothermic peak can be observed at around 170 °C. This corresponds to the glass transition temperature of the material. At around 343 °C, a larger, endothermic, peak is observed, corresponding to the melting temperature of the polymer. Two exothermic peaks are observed at temperatures corresponding to the decomposition of the material (580 °C). This confirms that the decomposition of PEEK is at least a two-step process when it is heated at 10 °C/min under nitrogen.

The enthalpy of decomposition measured was −417 J/g. Moreover, the exothermic nature of these peaks is unusual in the sense that common polymers show an endothermic peak during their decomposition. This can be explained by the fact that PEEK is a highly charring polymer [10]. This means that during its thermal decomposition, it forms a highly stable carbonaceous structure, implying that the heat released during bond formation in the charred structure is higher than that released during the bond breaking process during the decomposition.

This exothermic nature of the thermal decomposition of PEEK means that it releases heat when it degrades. While this property is not highly enviable in a fire scenario, it also suggests that the char structure that is formed during its initial decomposition is highly stable and can possibly act as a fire barrier, protecting the underlying material while the exposed surface decomposes.

Studying the thermal stability and behavior of PEEK under different oxygen concentrations has provided insight on the dependence of the oxygen concentration on the different steps of the thermal decomposition of PEEK. Indeed, whether a thermal stress is applied in the absence of oxygen or under air, the temperature at which the first step of decomposition occurs is only slightly affected. This implies that the reigning reaction that occurs as the polymer first starts to degrade and decompose thermally is the action of heat only. It is therefore of interest to study the nature of the decomposition gases that are evolved during the thermal decomposition of this material.

### 3.2. Identification of Decomposition Products of PEEK

#### 3.2.1. Pyrolysis Products of PEEK

To identify the products released during its pyrolysis, PEEK was subjected to TGA coupled with FTIR. TGA–FTIR of PEEK under nitrogen yielded many different pyrolysis products. The main decomposition products identified correspond to carbon dioxide and carbon monoxide. In order to qualitatively compare the gases evolved, intensity profiles corresponding to characteristic peaks of some degradation products (CO_2_ at 2356 cm^−1^, CO at 2184 cm^−1^ and CH_4_ at 3018 cm^−1^) were plotted (Figure 4).

The peak intensity of CO_2_ remains stable until around 30 min (approximately 530 °C). At this temperature, there is a negligible mass loss recorded on the TGA plot. After this, the peak intensity of CO_2_ increases gradually until it reaches a maximum at around 35 min (600 °C). It then decreases to a local minimum at 40 min (737 °C) before increasing again. These maxima/minima suggest that there may be more than two steps involved in the thermal decomposition mechanism of PEEK.

In the case of CO, the peak intensity profile shares a similar shape to that of CO_2_ at the beginning. However, as opposed to the CO_2_ intensity plot, the CO plot shows a second maximum peak at 42 min. This suggests that there may be different reactions producing CO at different temperatures.

From the intensity profile of methane, it can be seen that methane is released at a slightly higher temperature than CO and CO_2_. Indeed, the intensity of the methane peak starts to increase at 32 min (570 °C) and remains relatively constant until around 42 min. This means methane is released shortly after the main pyrolysis step (first decomposition) of PEEK. This can be explained by the secondary pyrolysis of volatile aromatics and the decomposition of the charred structure during the graphitization process (see mechanism described, Scheme 1, page 17) [12]. Moreover, the pyrolysis of a charred structure also leads to the formation of hydrogen gas, which can, in turn react with a carbonized structure to form methane [13]. Furthermore, secondary pyrolysis of volatile decomposition products (such as phenol) may also lead to methane formation [14,15,16].

In addition to the previously mentioned decomposition gases, other evolved gases were also identified. However, due to overlapping of some peaks in the FTIR spectra, characteristic peak profiles of the identified gases were not singled out. By comparing with reference FTIR spectra, other decomposition products were identified and are reported in Table 2. Of the decomposition gases, aromatic ethers, phenols, and furans were the main groups of products identified, suggesting a decomposition mechanism whereby the polymer chains are cut randomly.

The main decomposition products identified at 720 °C were methane, phenol, carbon dioxide and carbon monoxide. The products identified were previously reported in a study concerning the mechanism of the thermal decomposition of PEEK [12]. However, the release of methane was not reported.

In order to have better accuracy in the identification of evolved gases during the pyrolysis of PEEK, pyrolysis–gas chromatography–mass spectrometry (Py–GC–MS) was used. By subjecting a PEEK sample to the same heating rate as for the TGA–FTIR measurements, the evolved gases during its thermal decomposition are collected and separated via a gas chromatography column, which are then analyzed using a mass spectrometer. However, due to the nature of the instrument, the pyrolysis program can only be used in helium, therefore, this analysis “only” provides information regarding the pyrolysis products of the thermal decomposition of the material.

Figure 5 shows the intensity profiles of carbon dioxide and phenol, measured using EGA–MS method. Phenol (Figure 5, violet) has the highest intensity, followed by carbon dioxide (Figure 5, gold). This suggests that there is a significant amount of decarboxylation reactions occurring during the thermal decomposition of PEEK. Moreover, it has been reported that the release of CO_2_ is also associated with the pyrolytic decomposition of carbonyls [17]. The CO_2_ profile also evidences that there are two major steps occurring when PEEK is decomposing: two peaks are observed, one at 620 °C and the other at around 680 °C.

EGA–MS of PEEK at 20 °C/min shows that phenol has the intensity profile with the highest intensity (>150,000, Figure 5) than the other decomposition products (<15,000, Figure 6).

From the intensity profiles on Figure 5 and Figure 6, it can be observed that the onset of dibenzofuran release occurs at the lowest temperature (575 °C). CO_2_ is the next gas to be observed at around 600 °C. Most of the other decomposition gases start to be released within the temperature range 620–650 °C. The maximum intensities observed are within the range 640–670 °C.

#### 3.2.2. Thermal Decomposition Products of PEEK under Low Oxygen Concentration

In order to better understand the effect of oxygen on the thermal decomposition products released by PEEK, TGA–FTIR was performed with low (<2%) oxygen concentration. Figure 7 shows the intensity profiles of the different products that were identified during the TGA–FTIR analysis with 2% oxygen.

It can be observed on the peak intensity profiles that the highest intensity corresponds to that of CO_2_ peak. However, at the beginning of the decomposition (33 min), phenol, carbon monoxide, methane and 4-phenoxyphenol are also identified. Similar to the thermal decomposition of PEEK in nitrogen, the release of methane occurs at a slightly higher temperature than that of the release of phenol and carbon monoxide.

From the TGA, it can be deduced that the first step of the decomposition is not greatly influenced by the presence of oxygen. However, the second step, which corresponds to the degradation of the char formed, undergoes thermal oxidation when the atmosphere contains 2% oxygen. This is even clearer when comparing the CO_2_ intensities.

TGA–FTIR was also performed in a synthetic air atmosphere so as to get further information concerning its first step of degradation (Figure 8)

Only a few decomposition products were identified, as most pyrolysis products are probably thermo-oxidized before reaching the spectrometer. However, the presence of methane and phenol were still recorded… This means that at low temperatures (600 °C), despite the abundance of oxygen, these decomposition gases are not oxidized. This is coherent with our previous hypothesis: the first decomposition step has little or no dependence on the oxidative or inert nature of the atmosphere it is in and at the heating rate that was used.

From a fire behavior understanding, the decomposition products identified are potential flammable volatiles that can contribute to feeding a flame during a fire scenario. However, since a large proportion of the decomposition gases correspond to carbon dioxide (and possibly carbon monoxide), it is possible that dilution of flammables may occur in the gas phase when the material is subjected to heat. Indeed, this was evidenced by comparing the heat of combustion of PEEK to that of the gas phase decomposition products using a combustion flow calorimeter. The total heat of combustion of PEEK, measured using an oxygen bomb calorimeter [18] was found to be around 31 kJ/g, whereas the heat of combustion of its decomposition products, measured using a microscale combustion calorimeter summed up to around 12 kJ/g. This means that during a fire scenario, more than 50% of the heat released during the burning of PEEK comes from the thermal oxidation of the char structure and not from the decomposition gases themselves.

These results show that the thermal decomposition of PEEK leads to a plethora of decomposition products, whatever the atmosphere in which it is heated in. However, it can be hypothesized that in an oxidative environment, the gases released during the initial decomposition step are, for the most part, thermo-oxidized to form mostly carbon dioxide. This means whatever the atmosphere, the first reactions occurring would correspond to the formation of a char and the release of the decomposition gases. However, the fire behavior of the material will depend on the concentration of these decomposition gases as a material will only ignite if there is enough combustibles as well as oxygen availability. Moreover, the different stages of a fire scenario correspond to different kinds of thermal stress: before ignition (thermo-oxidation), transient stage (low oxygen concentration or oxygen deprived), during flaming combustion (oxygen deprived) and after flameout (thermo-oxidation).

### 3.3. Fire Behavior of PEEK

In an attempt at linking the decomposition behavior of PEEK in the previous section to its fire behavior, the latter has been studied using the mass loss cone calorimetry (MLC).

Three different incident heat fluxes were used: 35 kW/m², 50 kW/m² and 60 kW/m² (Figure 9) on PEEK plaques to study its fire behavior.

When subjected to an incident heat flux of 35 kW/m², PEEK does not ignite, even after 1000 s. Visual observations made included the formation of small bubbles on the surface of the material as well as slight melting.

When subjected to an incident heat flux of 50 kW/m², the HRR of the material remains zero until it ignites (843 s). After ignition, there is a sudden increase in HRR, which peaks at 125 kW/m² (990 s).

It is worth noting that, right before ignition, the material had swollen by more than 1000% (visual observations, Figure 10). This means that the gases released during the thermal decomposition of PEEK at this incident heat flux do not readily ignite, however, upon their release, they pull up the char formed. This protects the underlying polymeric material by preventing the incident heat flux from reaching it.

The swelling phenomenon can be observed on Figure 10, which corresponds to pictures taken at different times during the test. Indeed, well before ignition occurs, at 307 s, significant swelling is already observed. As the swelling increases, the material gets increasingly close to the heating resistance. This means that the actual incident heat flux reaching the material is higher than the one measured at the initial sample position. Moreover, it should be noted that the igniter (red circle, Figure 10) does not contribute to the ignition. In fact, the region below the igniter is less swollen, possibly because of the lower heat flux incident on that part of the material due to a shielding effect brought about by the presence of the igniter. Once the polymer ignited, the exposed surface smoldered (Figure 10, middle) before properly flaming up (Figure 10, right). It is interesting to note that the ignition occurs on the inside of the material.

In order to dig deeper in the reactions that may be occurring during the thermal decomposition of PEEK under the incident heat flux, an FTIR was connected to the exhaust to analyze the gases evolved during the whole test. This method allowed for the quantitative analysis of water, carbon dioxide and carbon monoxide evolved during the test. Figure 11 shows the HRR curve as well as the profiles of the aforementioned gases with respect to time during the MLC test.

From the intensity profile of water, it can be observed that there is no water released before ignition. During the burning process, the intensity curve of water has a similar shape as the HRR and CO_2_ curve, as it can be expected for a combustion scenario.

Furthermore, the intensity profile of CO and CO_2_ reveal that before ignition, there is a non-negligible amount of these gases that is produced. This means that the swelling that is observed during the charring phenomenon is partly led by the release of CO and CO_2_, among other decomposition gases.

At 60 kW/m², PEEK exhibits a quite different behavior than at 50 kW/m². The HRR curve concerning the burning behavior of PEEK can be split into two parts, the first one corresponds to the rapid increase of HRR from its ignition to the first peak of HRR (Figure 9). The second part, whereby the heat release rate starts to increase again and plateaus before slowly decreasing to zero.

Similar to the test carried out at 50 kW/m², HRR of PEEK when it is subjected to 60 kW/m² remains close to zero until ignition at 113 s. After ignition, the heat release rate increases rapidly until around 105 kW/m² before decreasing again. This decrease in the HRR may be explained by the limited supply of fuel by the polymer, due to the char structure that is formed during this first phase of burning which may have temporarily protected the inner layer of the material. However, this protective layer is quickly smoldered into ashes, and the fire on the polymer rises again. This is visible on the HRR curve whereby the HRR starts to increase again at 309 s. The peak HRR reached after this, is 107 kW/m². It remains relatively constant around this value until 480 s, where it starts to gradually decrease again until the material flames out.

From the pictures of PEEK taken just before ignition, some swelling can be observed (Figure 12, left). Fumes are escaping the bulk material on the left-hand side of the material just before ignition. These fumes are ignited, leading to the subsequent burning of the material. This ignition causes the whole polymer to start burning, causing a rapid increase in the HRR, which peaks at 105 kW/m² (Figure 12, middle). However, at this stage, as PEEK burns, significant swelling is still observed (Figure 12, right). This swelling corresponds to significant char formation reactions and is coupled with a slight decrease in the heat release rate.

The second part of the burning process of PEEK under 60 kW/m² is also interesting. Indeed, after a slight decrease in HRR at 150 s, the HRR starts to rise again. A relatively small flame remains present for around 3 min and subsides. During this time, the HRR stagnates at around 107 kW/m². After the attenuation of the flame, the heat release rate does not instantaneously reach a lower value but exhibits a gradual decrease. During this time, the material is in a smoldering state, whereby it slowly degrades, corresponding to the incandescence observed at *t* > 400 s. It means that the material is being thermo-oxidized due to the high incident heat flux. The degradation effect is clearly visible on the material (Figure 13, right) as there is hardly any of it left at the end of the test.

From the investigation of its fire behavior, PEEK has proved to exhibit excellent resistance to fire. This is in accordance with the high thermal stability that was observed in the previous section. Indeed, the first step that was observed when PEEK was subjected to an incident heat flux corresponded with the formation of a char. However, the swelling behavior was not predictable from the TGA analyses. There seems to a correlation between the thermal decomposition products and the charring behavior of the polymeric material that enhances the fire behavior of PEEK, especially at 50 kW/m². The next section attempts to elucidate this by determining the thermal decomposition pathway of the material when it is subjected to a thermal stress.

### 3.4. Thermal Decomposition Reactions of PEEK under Pyrolytic Conditions

From the STA of PEEK, a highly exothermic decomposition peak was observed (−417 J/g). This was attributed to the fact that a large energetic contribution goes to the formation of a stable char. In order to better understand this char formation, the decomposition mechanism of PEEK is investigated hereafter.

From the TGA analyses, it was deduced that the thermal decomposition process was at least a two-step one. This was confirmed by the TGA–FTIR and pyrolysis GC–MS whereby different sets of decomposition products were detected at different temperatures. Moreover, the thermal decomposition products evolved during the MLC test have also provided insight as to the different stages of thermal decomposition occurring in PEEK when it is subjected to an incident heat flux.

From the MLC–FTIR and TGA–FTIR results, it was observed that the first degradation products correspond to the formation of carbon monoxide and carbon dioxide.

The formation of carbon monoxide is related to the graphitization mechanism (Scheme 1). This corroborates with the MLC test as both swelling and charring were observed during the test (Figure 10, page 14). Previous reports have suggested that the charring process occurs at temperatures above 750 °C. However, in the case of the MLC test at 50 kW/m², charring was observed while the temperature at the back of the sample was at around 500 °C. This means that the temperature on the surface is higher than 500 °C. Indeed, at this temperature, random scission of the polymer chain may release aromatic radicals. These radicals, when reacting with adjacent benzene rings on a benzophenone moiety leads to the formation of a fluorenone-like structure. Succeeding this reaction, the release of CO leads an aromatic diradical, two (or more) of which can react together to form a crosslinked aromatic structure. Scheme 1 is a summary of this charring process [12].

Another possible pathway explaining the formation of carbon monoxide is illustrated in the first step of the decomposition mechanism in Scheme 2. Homolytic scission of the carbonyl between two aromatic phenyls leads to the formation of carbon monoxide and two aromatic radicals. These aromatic radicals may combine to contribute to the carbonization process described in Scheme 1.

Moreover, from the EGA–MS measurements, it was observed that dibenzofuran was one of the first components to be detected. This suggests that it is released relatively early during its degradation. However, a direct mechanism for its formation seems improbable. Indeed, these two molecules involve a furan ring, which is unlikely to form at high temperatures because decomposition products such as phenol or phenoxyphenol are much more likely to occur [12]. One suggested pathway which could explain the formation of dibenzofuran and dibenzofuranol involves a two-step process. One, which occurs at low temperature, leading to the formation of the furan moiety on the polymer backbone, and another whereby random scission of the backbone leads to the release of dibenzofuran and dibenzofuranol (Scheme 2). From the relative intensity profiles on the EGA–MS of PEEK, it can be deduced that the major product of such a decomposition is the dibenzofuran rather than dibenzofuranol. Indeed, dibenzofuranol is detected at a higher temperature with a lower relative intensity than dibenzofuran (Figure 6). As for the formation of phenoxydibenzofuran, a similar mechanism is probable, whereby the carbonyl–phenyl bond is cleaved rather than an ether–phenyl one (Scheme 2).

Moreover, another bond that is susceptible to be cleaved at high temperatures is the ether–aromatic one. Indeed, it would lead to the formation of a phenoxy radical that is stabilized by an aromatic ring. Therefore, following the release of CO, it is likely that the cleavage of the ether–aromatic linkage leads to the formation of phenol (Scheme 3).

However, when the cleavage occurs over two adjacent ether–phenyl bonds, it leads to the formation of a benzoquinone (diphenoxy radical mesomere shown in Scheme 3). This diradical can abstract two hydrogens to form a hydroquinone. During the pyrolysis GC–MS, both hydroquinone and benzoquinone were observed, suggesting that both reactions are occurring during the thermal decomposition.

To explain the formation of carbon dioxide, possible rearrangement reactions need to be considered. Indeed, during thermal decomposition, CO_2_ may be generated from the formation of carboxylic acid derivatives such as aromatic ester as described in Scheme 4. This can be initiated by the cleavage of a bond between an aromatic ring and a carbonyl function and an ether bond, which subsequently rearrange to form an ester. The latter can then undergo a decarboxylation, to release carbon dioxide and two parts of polymer chain radicals.

Furthermore, at around 620 °C, phenol and diphenylether were detected both by TGA–FTIR and by Py–GC–MS. The same two bonds as for the explanation for the formation of 4-phenoxyphenol are involved. However, the placement of the scission and the subsequent reactions occurring are not the same.

Moreover, a cleavage of two adjacent carbonyl–phenyl bonds would lead to the formation of 1,4-diphenoxybenzene diradical. It can abstract a hydrogen on its either side leading to the formation of 1,4-diphenoxybenzene [12].

Despite the extensive graphitization assumed during the thermal decomposition of PEEK, hydroxybenzophenone was also detected during EGA–MS. Its formation can be explained by the fact that at high temperature, two ether–phenyl bonds are cleaved in such a way that they form the hydroxybenzophenone diradical. By abstracting two hydrogens, volatile hydroxybenzophenone is formed.

## 4. Conclusions

The thermal stability investigation on PEEK under different oxygen concentrations have brought to light that the onset of the thermal decomposition of PEEK is mostly independent of the oxidative or inert nature of the atmosphere. This suggests that initial step of the decomposition may involve the same reactions. However, when looking at the decomposition products at the beginning of the decomposition under low oxygen concentrations, a higher relative amount of carbon dioxide is observed than inert atmosphere, meaning that thermo-oxidation of the decomposition products occurs in the presence of oxygen. This step probably corresponds to the formation of a charred structure. However, when the char formed is in the presence of oxygen at high temperature, it starts to degrade thermally, showing a greater effect on the second step of the thermal decomposition.

Understanding the thermal decomposition of PEEK was essential in explaining its enhanced fire behavior. First, the initial stage of its thermal decomposition leads to the formation of a highly stable charred structure. This structure has the ability of protecting the bulk of the polymer from incident heat flux. Adding to this, the combustible decomposition products released by PEEK is probably diluted by the high amount of carbon dioxide and carbon monoxide released during the early decomposition stages. Furthermore, the swelling behavior observed during the mass loss cone calorimetry test also plays a role in retarding the ignition of PEEK by, once again, protecting the underlying undecomposed polymer, leading to a slow release of combustibles.

This charring of PEEK was possible due to the presence of a high amount of aromatic structures in the backbone of the polymer. Further evidence from the mechanism of char formation shows that having adjacent phenoxy radicals (stabilized because of the oxygen connected to the aromatic ring) is a valuable property to look for if char formation is to be promoted.

However, this phenomenon is not observed to the same extent when the incident heat flux is increased. Indeed, under a higher heat flux (60 kW/m²), the release of combustibles occurs at a higher rate (higher temperature). There is less time for the polymer to start melting, form a char and for the released gases to rise and drive with them the surface of the polymer. This also means that the decomposition occurs faster and that the concentration of decomposition products is higher. Py–GC–MS and EGA–MS have allowed us to identify the decomposition products that are released at high temperatures, some of which are highly inflammable. It is probable that the concentration of these products becomes high enough to cause ignition. As ignition occurs, the decomposition mechanism of the material continues at a higher rate as there is little protection of the bulk material by the charred surface.

Despite catching fire when subjected to high incident heat flux, the heat release rate of PEEK remains relatively low (<125 kW/m² both with an incident heat flux of 50 and 60 kW/m²) which is an impressively low value for a wholly organic polymeric material.

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
