# Peer review of "A Case Study of Polyether Ether Ketone (I): Investigating the Thermal and Fire Behavior of a High-Performance Material"

_polymers, 2020, doi:10.3390/polym12081789_

Round 1

Reviewer 1 Report

The manuscript “A Case Study of Polyetheretherketone (I): Investigating the Thermal and Fire Behavior of a High-Performance Material” is an interesting though very broad paper reporting on the thermal and fire behavior of the high-performance, semi-crystalline thermoplastic polymer polyetheretherketone (PEEK). The authors examine PEEK via thermogravimetric analyses (TGA and TGA/DSC) and under different oxygen concentrations (0 ‑ 20 % in synthetic air). Furthermore, simultaneous thermal analysis (STA) is applied. The thermal decomposition mechanism of PEEK is then investigated by pyrolysis-gas-chromatography/mass-spectrometry (Py-GC/MS) and TGA-FTIR, and decomposition products of PEEK are depicted. Additionally, he fire behavior of PEEK is investigated by means of mass loss cone calorimetry (MLC), and the decomposition products of PEEK evolved under conditions of different external heat fluxes (35, 50, and 60 kW/m²) plus ignition are analysed with a coupled FTIR with a gas analyses cell. Those findings were compared by the authors with the results from the pyrolysis experiments in the coupling of TGA with FTIR and Py-GC/MS. I the last part of the manuscript, the authors try to elucidate the mechanisms for the formation of the depicted decomposition products.

This study is a rather expansive and detailed piece of work, in which the authors explain extensively their findings and the proposed causes and mechanisms. However, many conclusions which are drawn are self-evident and do not need the excessive repetitions which the authors apply in the different sub-chapters (e.g. the influence of oxygen on the – thermal - decomposition). Additionally, even for readers not too familiar with fire sciences and polymer flammability, the main decomposition products of polymers, like carbon dioxide, water, and carbon monoxide are well known and do not necessarily need excessive description and explanation the authors apply here to explain their findings. Furthermore, after careful reading, some more issues became obvious, which I would like to bring to the attention of the authors:

  1. Please give adequate literature evidence for your statements in the introduction, e. g. on the fire-related deaths in the EU per year, and their cause by use of flammable materials (polymers?) as stated.
  2. Line 75: The use of “we” should generally be avoided in scientific papers.
  3. Line 91: Did the authors use and investigate their samples, after grinding in liquid nitrogen, without prior drying in the TGA experiments? If not please indicate that this was done and how.
  4. Line 109ff: Please check your sentence/spelling and grammar.
  5. In Fig. 3: Please give TGA and DSC as experiments for the 1st and 2nd y-axis (only done for DSC).
  6. The whole manuscript suffers from (too) many repetitions, e. g. in chapters 3.1.1 (line 232 ff.) and 3.2.1 (line 407 ff. and line 399 ff.); the authors should definitely try to combine and aggregate their findings and explanations.
  7. The explanation for the (potential) formation of methane (line 326 and Scheme 1) is very speculative, and the literature given (Yan et al., Polymers (Basel) 2018, 10 (7) DOI:3390/polym1070729 does not support the proposed mechanism for PEEK. Please give adequate experimental and/or literature evidence for your proposal.
  8. 4 (line 328): It does not become clear to the reader what the dashed lines shall indicate? Please explain or leave them out. Additionally, please indicate the corresponding wavenumber in the IR experiment for the three products detected and shown here (CO, CO2, CH4). Is this (rather low-intensity) detection of methane really characteristic for methane? Please explain why. The caption of Fig. 4 should also include that the data were obtained with FTIR experiments.
  9. 5 (line 342): These IR spectra shown are of no use when the absorbance bands not assigned. The authors should – at least – assign their identified products (compare Table 2, line 347 ff.).
  10. The caption of Table 2 (line 347 ff) should give the full method (nitrogen atmosphere?), and absolutely necessarily give the main wavenumbers at which the authors detected the proposed compounds/structures in the FTIR experiment.
  11. Again, in Fig. 7 (line 365) as well the meaning of the dashed lines should be explained, as well as the method given (under nitrogen? With or without O2?) Additionally, the m/z, which should be given for the proposed decomposition products recorded in the EGA-MS experiment.
  12. In Fig. 8 (line 370) the temperature is most probably more meaningful than the time given here on the x-axis. But by far more important is, that for two of the proposed compounds the given masses do not match (1,4-benzochinone = 108, 2-phenoxydibenzo[b,d]furan = 198), which is decisive as an explanation for their (proposed) formation is given in Scheme 2 (line 586). Furthermore, the mass of the proposed biphenyl is not given at all.
  13. Literature evidence for the proposed MCC heat of combustion for PEEK (line 426) should be given.
  14. Details of the conducted experiment should be given in the caption of In Fig. 13 (line 484), e. g. at which irradiation it was performed (50 kW/m² ?)
  15. The authors should additionally note (and mention!) that not only the curve of water has the same shape as the HRR curve (line 487), but also the curve of the CO2 formation (as expected!).
  16. Line 519: “Curious” is not the right word here; should be “interesting”, or similar… Many more errors in grammar in 519 ff., which should be corrected.
  17. Many of the proposed decomposition mechanisms shown in chapter 3.4, especially those given in Scheme 3 and Scheme 4 to explain the formation of methane during PEEK decomposition, are highly speculative, and unfortunately the authors miss providing fundamental data supporting them (e.g. by quantum-mechanical calculations, etc.). Without any proof or data supporting these mechanisms, those highly speculative pathways should not be included in the manuscript. Overall, most of the proposed reaction pathways lack experimental or mathematical data supporting them, and should therefore urgently be reconsidered by the authors.

Author Response

The manuscript “A Case Study of Polyetheretherketone (I): Investigating the Thermal and Fire Behavior of a High-Performance Material” is an interesting though very broad paper reporting on the thermal and fire behavior of the high-performance, semi-crystalline thermoplastic polymer polyetheretherketone (PEEK). The authors examine PEEK via thermogravimetric analyses (TGA and TGA/DSC) and under different oxygen concentrations (0 ‑ 20 % in synthetic air). Furthermore, simultaneous thermal analysis (STA) is applied. The thermal decomposition mechanism of PEEK is then investigated by pyrolysis-gas-chromatography/mass-spectrometry (Py-GC/MS) and TGA-FTIR, and decomposition products of PEEK are depicted. Additionally, he fire behavior of PEEK is investigated by means of mass loss cone calorimetry (MLC), and the decomposition products of PEEK evolved under conditions of different external heat fluxes (35, 50, and 60 kW/m²) plus ignition are analysed with a coupled FTIR with a gas analyses cell. Those findings were compared by the authors with the results from the pyrolysis experiments in the coupling of TGA with FTIR and Py-GC/MS. I the last part of the manuscript, the authors try to elucidate the mechanisms for the formation of the depicted decomposition products.

This study is a rather expansive and detailed piece of work, in which the authors explain extensively their findings and the proposed causes and mechanisms. However, many conclusions which are drawn are self-evident and do not need the excessive repetitions which the authors apply in the different sub-chapters (e.g. the influence of oxygen on the – thermal - decomposition). Additionally, even for readers not too familiar with fire sciences and polymer flammability, the main decomposition products of polymers, like carbon dioxide, water, and carbon monoxide are well known and do not necessarily need excessive description and explanation the authors apply here to explain their findings. Furthermore, after careful reading, some more issues became obvious, which I would like to bring to the attention of the authors:

Thank you for these general remarks. We have attempted at making the explanations more concise. However, the obviousness of the influence of oxygen in the thermal decomposition is not as obvious, especially in high performance polymeric materials. Some materials (such as polybenzoxazole (zylon) and paraaramid (or Kevlar/Twaron)) show a large difference in decomposition temperature when the analysis is performed in air or under nitrogen. This is primarily why such emphasis was put in describing and explaining the thermal decomposition of PEEK, which has very little difference in thermal decomposition temperature in air as compared to that under nitrogen (see our review, “Thermal Degradation and Fire Behavior of High Performance Polymers”, Polymer Reviews 59(1) (2019) 55-123)

  1. Please give adequate literature evidence for your statements in the introduction, e. g. on the fire-related deaths in the EU per year, and their cause by use of flammable materials (polymers?) as stated.

The reference linked to this literature is the same as the sentence that follows. The data come from the official website of fire safe Europe. If the reviewer thinks that the official website does not fulfil the requirements to being an adequate literature, we will change this sentence.

  1. Line 75: The use of “we” should generally be avoided in scientific papers.

Thank you for this remark. The mistake has been corrected at that point and others throughout the text. Note it is a typically debate between British and American.

  1. Line 91: Did the authors use and investigate their samples, after grinding in liquid nitrogen, without prior drying in the TGA experiments? If not please indicate that this was done and how.

The samples were not dried before the TGA experiments.

  1. Line 109ff: Please check your sentence/spelling and grammar.

The sentence has been modified.

  1. In Fig. 3: Please give TGA and DSC as experiments for the 1st and 2nd y-axis (only done for DSC).

The figure has been modified with “TG” written on the left y-axis.

  1. The whole manuscript suffers from (too) many repetitions, e. g. in chapters 3.1.1 (line 232 ff.) and 3.2.1 (line 407 ff. and line 399 ff.); the authors should definitely try to combine and aggregate their findings and explanations.

The manuscript has been reviewed and condensed to make the explanations more concise.

  1. The explanation for the (potential) formation of methane (line 326 and Scheme 1) is very speculative, and the literature given (Yan et al., Polymers (Basel) 2018, 10 (7) DOI:3390/polym1070729 does not support the proposed mechanism for PEEK. Please give adequate experimental and/or literature evidence for your proposal.

Additional literature evidence has been added to support the proposal.

  1. 4 (line 328): It does not become clear to the reader what the dashed lines shall indicate? Please explain or leave them out. Additionally, please indicate the corresponding wavenumber in the IR experiment for the three products detected and shown here (CO, CO2, CH4). Is this (rather low-intensity) detection of methane really characteristic for methane? Please explain why. The caption of Fig. 4 should also include that the data were obtained with FTIR experiments.

Thank you for this remark. We have removed the dashed lines from the figure, the corresponding wavenumbers for the three products have been added in the caption, as well as the fact that the data were obtained from FTIR measurements.

As for the characteristic peak of methane, it is indeed that of methane. Please find a reference gas phase FTIR of methane below which shows a peak at 3018 cm-1.

  1. 5 (line 342): These IR spectra shown are of no use when the absorbance bands not assigned. The authors should – at least – assign their identified products (compare Table 2, line 347 ff.).

The identified products were assigned by comparing with reference FTIR spectra of the mentioned products. However, we can make the reference FTIR spectra available in supplementary data. They will all be like the one above, with each corresponding reference spectrum. Another solution would be to simply remove the two FTIR spectra and only leave the table, which we have done.

  1. The caption of Table 2 (line 347 ff) should give the full method (nitrogen atmosphere?), and absolutely necessarily give the main wavenumbers at which the authors detected the proposed compounds/structures in the FTIR experiment.

The specifics concerning the experimental procedure has been added. Moreover, to identify the different compounds, the FTIR spectra were compared with reference FTIR spectra from the NIST Chemistry WebBook. This was done in a similar manner as the comparison shown in the above example (with methane). The main wavenumbers corresponding to the different identified products have been added to the table 2.

  1. Again, in Fig. 7 (line 365) as well the meaning of the dashed lines should be explained, as well as the method given (under nitrogen? With or without O2?) Additionally, the m/z, which should be given for the proposed decomposition products recorded in the EGA-MS experiment.

The meaning of the dashed and filled lines have been added on the figure caption. The method has also been added on the figure caption (under helium)

  1. In Fig. 8 (line 370) the temperature is most probably more meaningful than the time given here on the x-axis. But by far more important is, that for two of the proposed compounds the given masses do not match (1,4-benzochinone = 108, 2-phenoxydibenzo[b,d]furan = 198), which is decisive as an explanation for their (proposed) formation is given in Scheme 2 (line 586). Furthermore, the mass of the proposed biphenyl is not given at all.

Time and temperature are both on the X-axes (one at the bottom, and the other on top). The mistake concerning the mass of benzoquinone has been corrected. Concerning the m/z of 2-phenoxydibenzo[b,d]furan, the m/z that was taken corresponds to the m/z fragment with the highest intensity in the MS and not to the molar mass of the molecule. In most cases the m/z values to correspond to the m/z of the molecules, but not in this case.

  1. Literature evidence for the proposed MCC heat of combustion for PEEK (line 426) should be given.

The evidence comes from measurements that was performed at our laboratory.

  1. Details of the conducted experiment should be given in the caption of In Fig. 13 (line 484), e. g. at which irradiation it was performed (50 kW/m² ?)

The details have been added to the caption.

  1. The authors should additionally note (and mention!) that not only the curve of water has the same shape as the HRR curve (line 487), but also the curve of the CO2 formation (as expected!).

Thank you for this remark. The details have been added

  1. Line 519: “Curious” is not the right word here; should be “interesting”, or similar… Many more errors in grammar in 519 ff., which should be corrected.

Thank you. The paragraph has been corrected.

  1. Many of the proposed decomposition mechanisms shown in chapter 3.4, especially those given in Scheme 3 and Scheme 4 to explain the formation of methane during PEEK decomposition, are highly speculative, and unfortunately the authors miss providing fundamental data supporting them (e.g. by quantum-mechanical calculations, etc.). Without any proof or data supporting these mechanisms, those highly speculative pathways should not be included in the manuscript. Overall, most of the proposed reaction pathways lack experimental or mathematical data supporting them, and should therefore urgently be reconsidered by the authors.

The mechanism explaining the formation of methane was adapted from a study of a model decomposition compound: diphenyl ether. A sentence justifying the proposed attempted mechanism has been added:

“These schemes are adapted principally by the study of the pyrolysis behavior of diphenyl ether, which evidenced the release of naphthalene and cyclopentadiene as decomposition products [19]. Based on this work, attempts at finding the most viable pathway explaining the carbonization of PEEK, to form “naphthalene” moieties by starting with the ether linkage.”

As for quantum mechanical calculations, we have not performed them as they do not form part of primary the goal of this work.

Reviewer 2 Report

The authors should revise the manuscript. Followings are my suggestion;

  1. The abstract and conclusion should at same agreement of findings, besides, the conclusion is difficult to follow. 
  2. The written content should be improved, i just to avoid too much of helping verb. Also, thankfully, thanks to, thank etc. the manuscript should portrait scientific findings.
  3. a lot of decomposition reactions are given, mostly proposed ones from literature. Is the author think it helps in burning behavior or gas emission, if its not focusing on pyrolysis of the material? The char has some specific use ?
  4. the literature review is low, it needs to be improved. 

Author Response

The manuscript “A Case Study of Polyetheretherketone (I): Investigating the Thermal and Fire Behavior of a High-Performance Material” is an interesting though very broad paper reporting on the thermal and fire behavior of the high-performance, semi-crystalline thermoplastic polymer polyetheretherketone (PEEK). The authors examine PEEK via thermogravimetric analyses (TGA and TGA/DSC) and under different oxygen concentrations (0 ‑ 20 % in synthetic air). Furthermore, simultaneous thermal analysis (STA) is applied. The thermal decomposition mechanism of PEEK is then investigated by pyrolysis-gas-chromatography/mass-spectrometry (Py-GC/MS) and TGA-FTIR, and decomposition products of PEEK are depicted. Additionally, he fire behavior of PEEK is investigated by means of mass loss cone calorimetry (MLC), and the decomposition products of PEEK evolved under conditions of different external heat fluxes (35, 50, and 60 kW/m²) plus ignition are analysed with a coupled FTIR with a gas analyses cell. Those findings were compared by the authors with the results from the pyrolysis experiments in the coupling of TGA with FTIR and Py-GC/MS. I the last part of the manuscript, the authors try to elucidate the mechanisms for the formation of the depicted decomposition products.

This study is a rather expansive and detailed piece of work, in which the authors explain extensively their findings and the proposed causes and mechanisms. However, many conclusions which are drawn are self-evident and do not need the excessive repetitions which the authors apply in the different sub-chapters (e.g. the influence of oxygen on the – thermal - decomposition). Additionally, even for readers not too familiar with fire sciences and polymer flammability, the main decomposition products of polymers, like carbon dioxide, water, and carbon monoxide are well known and do not necessarily need excessive description and explanation the authors apply here to explain their findings. Furthermore, after careful reading, some more issues became obvious, which I would like to bring to the attention of the authors:

Thank you for these general remarks. We have attempted at making the explanations more concise. However, the obviousness of the influence of oxygen in the thermal decomposition is not as obvious, especially in high performance polymeric materials. Some materials (such as polybenzoxazole (zylon) and paraaramid (or Kevlar/Twaron)) show a large difference in decomposition temperature when the analysis is performed in air or under nitrogen. This is primarily why such emphasis was put in describing and explaining the thermal decomposition of PEEK, which has very little difference in thermal decomposition temperature in air as compared to that under nitrogen (see our review, “Thermal Degradation and Fire Behavior of High Performance Polymers”, Polymer Reviews 59(1) (2019) 55-123)

  1. Please give adequate literature evidence for your statements in the introduction, e. g. on the fire-related deaths in the EU per year, and their cause by use of flammable materials (polymers?) as stated.

The reference linked to this literature is the same as the sentence that follows. The data come from the official website of fire safe Europe. If the reviewer thinks that the official website does not fulfil the requirements to being an adequate literature, we will change this sentence.

  1. Line 75: The use of “we” should generally be avoided in scientific papers.

Thank you for this remark. The mistake has been corrected at that point and others throughout the text. Note it is a typically debate between British and American.

  1. Line 91: Did the authors use and investigate their samples, after grinding in liquid nitrogen, without prior drying in the TGA experiments? If not please indicate that this was done and how.

The samples were not dried before the TGA experiments.

  1. Line 109ff: Please check your sentence/spelling and grammar.

The sentence has been modified.

  1. In Fig. 3: Please give TGA and DSC as experiments for the 1st and 2nd y-axis (only done for DSC).

The figure has been modified with “TG” written on the left y-axis.

  1. The whole manuscript suffers from (too) many repetitions, e. g. in chapters 3.1.1 (line 232 ff.) and 3.2.1 (line 407 ff. and line 399 ff.); the authors should definitely try to combine and aggregate their findings and explanations.

The manuscript has been reviewed and condensed to make the explanations more concise.

  1. The explanation for the (potential) formation of methane (line 326 and Scheme 1) is very speculative, and the literature given (Yan et al., Polymers (Basel) 2018, 10 (7) DOI:3390/polym1070729 does not support the proposed mechanism for PEEK. Please give adequate experimental and/or literature evidence for your proposal.

Additional literature evidence has been added to support the proposal.

  1. 4 (line 328): It does not become clear to the reader what the dashed lines shall indicate? Please explain or leave them out. Additionally, please indicate the corresponding wavenumber in the IR experiment for the three products detected and shown here (CO, CO2, CH4). Is this (rather low-intensity) detection of methane really characteristic for methane? Please explain why. The caption of Fig. 4 should also include that the data were obtained with FTIR experiments.

Thank you for this remark. We have removed the dashed lines from the figure, the corresponding wavenumbers for the three products have been added in the caption, as well as the fact that the data were obtained from FTIR measurements.

As for the characteristic peak of methane, it is indeed that of methane. Please find a reference gas phase FTIR of methane below which shows a peak at 3018 cm-1.

  1. 5 (line 342): These IR spectra shown are of no use when the absorbance bands not assigned. The authors should – at least – assign their identified products (compare Table 2, line 347 ff.).

The identified products were assigned by comparing with reference FTIR spectra of the mentioned products. However, we can make the reference FTIR spectra available in supplementary data. They will all be like the one above, with each corresponding reference spectrum. Another solution would be to simply remove the two FTIR spectra and only leave the table, which we have done.

  1. The caption of Table 2 (line 347 ff) should give the full method (nitrogen atmosphere?), and absolutely necessarily give the main wavenumbers at which the authors detected the proposed compounds/structures in the FTIR experiment.

The specifics concerning the experimental procedure has been added. Moreover, to identify the different compounds, the FTIR spectra were compared with reference FTIR spectra from the NIST Chemistry WebBook. This was done in a similar manner as the comparison shown in the above example (with methane). The main wavenumbers corresponding to the different identified products have been added to the table 2.

  1. Again, in Fig. 7 (line 365) as well the meaning of the dashed lines should be explained, as well as the method given (under nitrogen? With or without O2?) Additionally, the m/z, which should be given for the proposed decomposition products recorded in the EGA-MS experiment.

The meaning of the dashed and filled lines have been added on the figure caption. The method has also been added on the figure caption (under helium)

  1. In Fig. 8 (line 370) the temperature is most probably more meaningful than the time given here on the x-axis. But by far more important is, that for two of the proposed compounds the given masses do not match (1,4-benzochinone = 108, 2-phenoxydibenzo[b,d]furan = 198), which is decisive as an explanation for their (proposed) formation is given in Scheme 2 (line 586). Furthermore, the mass of the proposed biphenyl is not given at all.

Time and temperature are both on the X-axes (one at the bottom, and the other on top). The mistake concerning the mass of benzoquinone has been corrected. Concerning the m/z of 2-phenoxydibenzo[b,d]furan, the m/z that was taken corresponds to the m/z fragment with the highest intensity in the MS and not to the molar mass of the molecule. In most cases the m/z values to correspond to the m/z of the molecules, but not in this case.

  1. Literature evidence for the proposed MCC heat of combustion for PEEK (line 426) should be given.

The evidence comes from measurements that was performed at our laboratory.

  1. Details of the conducted experiment should be given in the caption of In Fig. 13 (line 484), e. g. at which irradiation it was performed (50 kW/m² ?)

The details have been added to the caption.

  1. The authors should additionally note (and mention!) that not only the curve of water has the same shape as the HRR curve (line 487), but also the curve of the CO2 formation (as expected!).

Thank you for this remark. The details have been added

  1. Line 519: “Curious” is not the right word here; should be “interesting”, or similar… Many more errors in grammar in 519 ff., which should be corrected.

Thank you. The paragraph has been corrected.

  1. Many of the proposed decomposition mechanisms shown in chapter 3.4, especially those given in Scheme 3 and Scheme 4 to explain the formation of methane during PEEK decomposition, are highly speculative, and unfortunately the authors miss providing fundamental data supporting them (e.g. by quantum-mechanical calculations, etc.). Without any proof or data supporting these mechanisms, those highly speculative pathways should not be included in the manuscript. Overall, most of the proposed reaction pathways lack experimental or mathematical data supporting them, and should therefore urgently be reconsidered by the authors.

The mechanism explaining the formation of methane was adapted from a study of a model decomposition compound: diphenyl ether. A sentence justifying the proposed attempted mechanism has been added:

“These schemes are adapted principally by the study of the pyrolysis behavior of diphenyl ether, which evidenced the release of naphthalene and cyclopentadiene as decomposition products [19]. Based on this work, attempts at finding the most viable pathway explaining the carbonization of PEEK, to form “naphthalene” moieties by starting with the ether linkage.”

As for quantum mechanical calculations, we have not performed them as they do not form part of primary the goal of this work.

The manuscript “A Case Study of Polyetheretherketone (I): Investigating the Thermal and Fire Behavior of a High-Performance Material” is an interesting though very broad paper reporting on the thermal and fire behavior of the high-performance, semi-crystalline thermoplastic polymer polyetheretherketone (PEEK). The authors examine PEEK via thermogravimetric analyses (TGA and TGA/DSC) and under different oxygen concentrations (0 ‑ 20 % in synthetic air). Furthermore, simultaneous thermal analysis (STA) is applied. The thermal decomposition mechanism of PEEK is then investigated by pyrolysis-gas-chromatography/mass-spectrometry (Py-GC/MS) and TGA-FTIR, and decomposition products of PEEK are depicted. Additionally, he fire behavior of PEEK is investigated by means of mass loss cone calorimetry (MLC), and the decomposition products of PEEK evolved under conditions of different external heat fluxes (35, 50, and 60 kW/m²) plus ignition are analysed with a coupled FTIR with a gas analyses cell. Those findings were compared by the authors with the results from the pyrolysis experiments in the coupling of TGA with FTIR and Py-GC/MS. I the last part of the manuscript, the authors try to elucidate the mechanisms for the formation of the depicted decomposition products.

This study is a rather expansive and detailed piece of work, in which the authors explain extensively their findings and the proposed causes and mechanisms. However, many conclusions which are drawn are self-evident and do not need the excessive repetitions which the authors apply in the different sub-chapters (e.g. the influence of oxygen on the – thermal - decomposition). Additionally, even for readers not too familiar with fire sciences and polymer flammability, the main decomposition products of polymers, like carbon dioxide, water, and carbon monoxide are well known and do not necessarily need excessive description and explanation the authors apply here to explain their findings. Furthermore, after careful reading, some more issues became obvious, which I would like to bring to the attention of the authors:

Thank you for these general remarks. We have attempted at making the explanations more concise. However, the obviousness of the influence of oxygen in the thermal decomposition is not as obvious, especially in high performance polymeric materials. Some materials (such as polybenzoxazole (zylon) and paraaramid (or Kevlar/Twaron)) show a large difference in decomposition temperature when the analysis is performed in air or under nitrogen. This is primarily why such emphasis was put in describing and explaining the thermal decomposition of PEEK, which has very little difference in thermal decomposition temperature in air as compared to that under nitrogen (see our review, “Thermal Degradation and Fire Behavior of High Performance Polymers”, Polymer Reviews 59(1) (2019) 55-123)

  1. Please give adequate literature evidence for your statements in the introduction, e. g. on the fire-related deaths in the EU per year, and their cause by use of flammable materials (polymers?) as stated.

The reference linked to this literature is the same as the sentence that follows. The data come from the official website of fire safe Europe. If the reviewer thinks that the official website does not fulfil the requirements to being an adequate literature, we will change this sentence.

  1. Line 75: The use of “we” should generally be avoided in scientific papers.

Thank you for this remark. The mistake has been corrected at that point and others throughout the text. Note it is a typically debate between British and American.

  1. Line 91: Did the authors use and investigate their samples, after grinding in liquid nitrogen, without prior drying in the TGA experiments? If not please indicate that this was done and how.

The samples were not dried before the TGA experiments.

  1. Line 109ff: Please check your sentence/spelling and grammar.

The sentence has been modified.

  1. In Fig. 3: Please give TGA and DSC as experiments for the 1st and 2nd y-axis (only done for DSC).

The figure has been modified with “TG” written on the left y-axis.

  1. The whole manuscript suffers from (too) many repetitions, e. g. in chapters 3.1.1 (line 232 ff.) and 3.2.1 (line 407 ff. and line 399 ff.); the authors should definitely try to combine and aggregate their findings and explanations.

The manuscript has been reviewed and condensed to make the explanations more concise.

  1. The explanation for the (potential) formation of methane (line 326 and Scheme 1) is very speculative, and the literature given (Yan et al., Polymers (Basel) 2018, 10 (7) DOI:3390/polym1070729 does not support the proposed mechanism for PEEK. Please give adequate experimental and/or literature evidence for your proposal.

Additional literature evidence has been added to support the proposal.

  1. 4 (line 328): It does not become clear to the reader what the dashed lines shall indicate? Please explain or leave them out. Additionally, please indicate the corresponding wavenumber in the IR experiment for the three products detected and shown here (CO, CO2, CH4). Is this (rather low-intensity) detection of methane really characteristic for methane? Please explain why. The caption of Fig. 4 should also include that the data were obtained with FTIR experiments.

Thank you for this remark. We have removed the dashed lines from the figure, the corresponding wavenumbers for the three products have been added in the caption, as well as the fact that the data were obtained from FTIR measurements.

As for the characteristic peak of methane, it is indeed that of methane. Please find a reference gas phase FTIR of methane below which shows a peak at 3018 cm-1.

  1. 5 (line 342): These IR spectra shown are of no use when the absorbance bands not assigned. The authors should – at least – assign their identified products (compare Table 2, line 347 ff.).

The identified products were assigned by comparing with reference FTIR spectra of the mentioned products. However, we can make the reference FTIR spectra available in supplementary data. They will all be like the one above, with each corresponding reference spectrum. Another solution would be to simply remove the two FTIR spectra and only leave the table, which we have done.

  1. The caption of Table 2 (line 347 ff) should give the full method (nitrogen atmosphere?), and absolutely necessarily give the main wavenumbers at which the authors detected the proposed compounds/structures in the FTIR experiment.

The specifics concerning the experimental procedure has been added. Moreover, to identify the different compounds, the FTIR spectra were compared with reference FTIR spectra from the NIST Chemistry WebBook. This was done in a similar manner as the comparison shown in the above example (with methane). The main wavenumbers corresponding to the different identified products have been added to the table 2.

  1. Again, in Fig. 7 (line 365) as well the meaning of the dashed lines should be explained, as well as the method given (under nitrogen? With or without O2?) Additionally, the m/z, which should be given for the proposed decomposition products recorded in the EGA-MS experiment.

The meaning of the dashed and filled lines have been added on the figure caption. The method has also been added on the figure caption (under helium)

  1. In Fig. 8 (line 370) the temperature is most probably more meaningful than the time given here on the x-axis. But by far more important is, that for two of the proposed compounds the given masses do not match (1,4-benzochinone = 108, 2-phenoxydibenzo[b,d]furan = 198), which is decisive as an explanation for their (proposed) formation is given in Scheme 2 (line 586). Furthermore, the mass of the proposed biphenyl is not given at all.

Time and temperature are both on the X-axes (one at the bottom, and the other on top). The mistake concerning the mass of benzoquinone has been corrected. Concerning the m/z of 2-phenoxydibenzo[b,d]furan, the m/z that was taken corresponds to the m/z fragment with the highest intensity in the MS and not to the molar mass of the molecule. In most cases the m/z values to correspond to the m/z of the molecules, but not in this case.

  1. Literature evidence for the proposed MCC heat of combustion for PEEK (line 426) should be given.

The evidence comes from measurements that was performed at our laboratory.

  1. Details of the conducted experiment should be given in the caption of In Fig. 13 (line 484), e. g. at which irradiation it was performed (50 kW/m² ?)

The details have been added to the caption.

  1. The authors should additionally note (and mention!) that not only the curve of water has the same shape as the HRR curve (line 487), but also the curve of the CO2 formation (as expected!).

Thank you for this remark. The details have been added

  1. Line 519: “Curious” is not the right word here; should be “interesting”, or similar… Many more errors in grammar in 519 ff., which should be corrected.

Thank you. The paragraph has been corrected.

  1. Many of the proposed decomposition mechanisms shown in chapter 3.4, especially those given in Scheme 3 and Scheme 4 to explain the formation of methane during PEEK decomposition, are highly speculative, and unfortunately the authors miss providing fundamental data supporting them (e.g. by quantum-mechanical calculations, etc.). Without any proof or data supporting these mechanisms, those highly speculative pathways should not be included in the manuscript. Overall, most of the proposed reaction pathways lack experimental or mathematical data supporting them, and should therefore urgently be reconsidered by the authors.

The mechanism explaining the formation of methane was adapted from a study of a model decomposition compound: diphenyl ether. A sentence justifying the proposed attempted mechanism has been added:

“These schemes are adapted principally by the study of the pyrolysis behavior of diphenyl ether, which evidenced the release of naphthalene and cyclopentadiene as decomposition products [19]. Based on this work, attempts at finding the most viable pathway explaining the carbonization of PEEK, to form “naphthalene” moieties by starting with the ether linkage.”

As for quantum mechanical calculations, we have not performed them as they do not form part of primary the goal of this work.

Round 2

Reviewer 1 Report

The revision reads better. Nevertheless, I still have the following issues for the authors to address properly.

I have read the revised manuscript carefully and found its content interesting although not quite ready (to my mind) for publication. Overall the authors have taken the previous reviewer comments seriously and have certainly improved the work: The applied methods are now described properly, the diagrams improved, and some grammar and/or spelling mistakes have been corrected.

However, some old concerns remain:

The manuscript reads very “raw” in pieces as it (a) still contains too many repetitions and trivialities in the whole text, (b) serious mistakes have not fully been corrected, e.g. still some masses in Fig. 6 do NOT match à 2-phenoxydibenzo[b,d]furan has 262 and NOT 260 (!), and (c) completely hypothetic reaction pathways without any proof are still included in the manuscript (scheme 3 and scheme 4) à they should be deleted from the manuscript or proper proof should be given. Also, just moving text blocks in the manuscript does not help to improve the quality. At last, especially the conclusion reads raw and trivial in pieces and can definitely be specified and improved.

I am fully aware of the pressures on authors to publish fast, and hence my recommendation is to accept after another revision, but I do not know if this recommendation is of any value. On the other hand, I am very sure that the authors can significantly improve their work by merely leaving out speculations and concentrating on and explaining their facts and findings. I am really looking forward to read a sincerely revised version of this manuscript, which will then without any doubt be of interest to the flame retardant community.

Author Response

Reviewer 1:

The revision reads better. Nevertheless, I still have the following issues for the authors to address properly.

I have read the revised manuscript carefully and found its content interesting although not quite ready (to my mind) for publication. Overall the authors have taken the previous reviewer comments seriously and have certainly improved the work: The applied methods are now described properly, the diagrams improved, and some grammar and/or spelling mistakes have been corrected.

However, some old concerns remain:

The manuscript reads very “raw” in pieces as it (a) still contains too many repetitions and trivialities in the whole text,

Significant effort has been made to reduce repetitions and trivialities (see the revised version of our manuscript).

 (b) serious mistakes have not fully been corrected, e.g. still some masses in Fig. 6 do NOT match à 2-phenoxydibenzo[b,d]furan has 262 and NOT 260 (!),

We believe that there may be a slight confusion between 2-phenoxydibenzo[b,d]furan and 1,4-diphenoxybenzene, both of which are present in the figure.

Number of atoms

Weight

Number of atoms

Weight

C

18

216

C

18

216

H

12

12

H

14

14

O

2

32

O

2

32

Total

260

262

and (c) completely hypothetic reaction pathways without any proof are still included in the manuscript (scheme 3 and scheme 4) à they should be deleted from the manuscript or proper proof should be given.

While the reaction may appear to be greatly hypothetic, they are not. The different intermediate products formed were inspired from the thermal decomposition pathway of phenol and diphenyl ether.

Seeing how the ether-phenyl bond can be broken relatively “easily”, it makes sense to take phenol and diphenylether moiety as model compounds to explain the decomposition of PEEK. For example, the tautomerization of phenol, followed by the release of carbon monoxide to give pentadiene [1] can be intuitively transposed to decomposition of the diphenylether moiety. We believe that basing oneself on the thermal decomposition of phenol is consistent with the chemical structure of the polymer, especially given the known fact that PEEK is a highly charring polymer, meaning that there is probably different pathways that lead to a polycyclic carbonized structure. Our reasoning of the use of phenol and diethylether as model compounds are given in the scheme below:

The literature reference to this reasoning is given in ref [1].

 Also, just moving text blocks in the manuscript does not help to improve the quality. At last, especially the conclusion reads raw and trivial in pieces and can definitely be specified and improved.

As mentioned, we have made significant effort to improve the readability and conciseness of the whole of our manuscript (see the revised version of our manuscript).

I am fully aware of the pressures on authors to publish fast, and hence my recommendation is to accept after another revision, but I do not know if this recommendation is of any value. On the other hand, I am very sure that the authors can significantly improve their work by merely leaving out speculations and concentrating on and explaining their facts and findings. I am really looking forward to read a sincerely revised version of this manuscript, which will then without any doubt be of interest to the flame retardant community.

Thank you for these comments, should the reviewer wish to discuss further, we would be happy to oblige.

  1. Scheer, A. M. Thermal Decomposition Mechanisms of Lignin Model Compounds: From Phenol to Vanillin, University of Colorado, 2011.

Reviewer 2:

The reviewer suggests extensive editing of English language. However, we believe that the level of English is more than proper. We have revised our manuscript and scrutinized eventual English language issues.

Reviewer 2 Report

It seems the authors have modified the manuscript but haven’t provide any scientific backing to their findings.

What is the rationale of self-claimed or hypothesis presented from line 211- onward? I suggest, the authors should provide references for fire scenarios posed in this section.

Again, at line 459, “it can be deduced that the first step of the decomposition is not greatly influenced by the presence of oxygen.” No reference was provided. Strong argument with their finding and literature should be carried out.

Author Response

It seems the authors have modified the manuscript but haven’t provide any scientific backing to their findings.

What is the rationale of self-claimed or hypothesis presented from line 211- onward? I suggest, the authors should provide references for fire scenarios posed in this section.

Response: The reference number 11 contains a mistake. It should have been: Witkowski A, Stec AA, Hull TR. Thermal Decomposition of Polymeric Materials. In: SFPE Handbook of Fire Protection Engineering. Springer New York; 2016:167-254. doi:10.1007/978-1-4939-2565-0_7

This literature reference, in our opinion provides adequate information regarding the effect of the thermal decomposition of a polymer on its fire behaviour. We have extracted some sentences from many others which evidence the appropriateness of this reference:

On the appropriateness of studying thermal decomposition: “Flaming combustion requires the fuel to be present in molecular form in the vapour phase, where it can undergo much more rapid reaction with atmospheric oxygen. Since polymers are much too large to exist in the vapour phase (because the bonding forces holding them in the condensed phase is proportional to their large surface area) they must first break down into volatile fragments.”

On thermal decomposition and oxygen: “At higher temperatures the majority of the bonds reach failure point, causing the release of gaseous molecules which differ depending on the material burning. This can be accelerated by attack of oxygen on the surface of the polymer, producing carbon dioxide and carbon monoxide.”

On the influence of oxygen: “In many polymers, the thermal decomposition processes are accelerated by oxygen, lowering the minimum decomposition temperatures”

Again, at line 459, “it can be deduced that the first step of the decomposition is not greatly influenced by the presence of oxygen.” No reference was provided. Strong argument with their finding and literature should be carried out.

Response: Thank you for pointing out this sentence. We have modified the sentence in order to bring more clarity to the mentioned observation.

The deduction made at this point involves the comparison between the temperature at the onset of the decomposition of PEEK under nitrogen (inert, no oxygen) with that under 2% oxygen. Since the sentence involves the comparison between two experimental results, one of which, is inexistent in the literature (TGA of PEEK under 2% oxygen), we believe that adding a reference here would be futile and misleading.

Round 3

Reviewer 1 Report

Thank you for giving additional and decisive literature evidence to support your findings. Thank you even more for specifying your conclusion, which really improves the manuscript.

I am sorry to have been confused with the EGA-MS results in figure 6 - you are right! The different positions for rather similar masses (260 vs 262) on the y-achses (m/z) in the diagram are not intuitive. Perhaps this visualization can still be improved?

The proposed mechanisms, especially for the release of methane, in schemes 3 and 4 are not convincing and, even more important as the authors themselves now give literature evidence for such formation from phenols. This should be sufficient and does not need excessive description in two(!) schemes which do not deliver any additional merit. I personally still do not like too much the suggestions given in schemes 3 and 4, but this is certainly highly subjective.

Author Response

Thank you for giving additional and decisive literature evidence to support your findings. Thank you even more for specifying your conclusion, which really improves the manuscript.

I am sorry to have been confused with the EGA-MS results in figure 6 - you are right! The different positions for rather similar masses (260 vs 262) on the y-achses (m/z) in the diagram are not intuitive. Perhaps this visualization can still be improved?

Because of the three-dimensional nature of the figure, we have chosen to arrange the positions in such a way that no curve is hidden by the one before it. If the reviewer accepts, we would like to keep the same arrangement.

The proposed mechanisms, especially for the release of methane, in schemes 3 and 4 are not convincing and, even more important as the authors themselves now give literature evidence for such formation from phenols. This should be sufficient and does not need excessive description in two(!) schemes which do not deliver any additional merit. I personally still do not like too much the suggestions given in schemes 3 and 4, but this is certainly highly subjective.

We agree with the reviewer and have therefore removed the schemes 3 and 4 as well as their corresponding details in the manuscript.

Reviewer 2 Report

I have no further comment.

Author Response

Reviewer 2:

The reviewer suggests extensive editing of English language. However, we believe that the level of English is more than proper. We have revised our manuscript and scrutinized eventual English language issues.